# Polyphenolic Hispolon Derived from Medicinal Mushrooms of the *Inonotus* and *Phellinus* Genera Promotes Wound Healing in Hyperglycemia-Induced Impairments

**DOI:** 10.3390/nu17020266

**Published:** 2025-01-13

**Authors:** Yi-Shan Liu, Mei-Chou Lai, Yu-Cheng Tzeng, I-Min Liu

**Affiliations:** 1Department of Dermatology, E-Da Hospital, I-Shou University, Kaohsiung City 84001, Taiwan; 2School of Chinese Medicine for Post Baccalaureate, College of Medicine, I-Shou University, Kaohsiung City 84001, Taiwan; 3Department of Pharmacy and Master Program, Collage of Pharmacy and Health Care, Tajen University, Yanpu Township 90741, Taiwan; meei@tajen.edu.tw; 4Kaohsiung Medical University Hospital, Kaohsiung Medical University, Kaohsiung City 80708, Taiwan; hahahahanelson@gmail.com

**Keywords:** wound healing, high glucose, hispolon, macrophage polarization, collagen synthesis

## Abstract

**Background**: This study investigated the wound-healing potential of hispolon, a polyphenolic pigment derived from medicinal mushrooms, under diabetic conditions using both in vitro and in vivo models. **Methods**: In the in vitro assays, L929 fibroblast cells exposed to high glucose (33 mmol/L) were treated with hispolon at concentrations of 2.5, 5, 7.5, or 10 μmol/L. In the in vivo assays, streptozotocin-induced diabetic rats with excision wounds received daily topical applications of 0.2 g of 5% (*w*/*w*) hispolon ointment. **Results**: Hispolon improved cell viability; suppressed oxidative stress by reducing reactive oxygen species, lipid peroxidation, and oxidative DNA damage; and restored the reduced glutathione/oxidized glutathione ratio. The scratch assay demonstrated that hispolon at 10 μmol/L enhanced fibroblast migration impaired by high-glucose conditions. Treatment with 5% (*w*/*w*) hispolon ointment accelerated wound contraction, reduced the epithelialization time, and enhanced tissue regeneration with an efficacy comparable to that of Fespixon^®^ cream, as shown by histological findings of increased fibroblast activity, collagen deposition, and capillary growth. Hispolon ointment also modulated macrophage polarization in diabetic wounds by reducing M1 markers and enhancing M2 markers. In a diabetic rat dead-space-wound model, 5% (*w*/*w*) hispolon ointment reduced the levels of pro-inflammatory cytokines, increased those of anti-inflammatory cytokines and growth factors, and stimulated Type I and III collagen synthesis, effectively promoting wound healing. In incisional wounds, hispolon ointment improved the wound-breaking strength, showing results comparable to that of Fespixon^®^ cream. Safety assessments confirmed that hispolon ointment showed no acute dermal toxicity. These findings underscore hispolon’s potential as a promising candidate for diabetic wound management by mitigating oxidative stress, enhancing tissue regeneration, and accelerating wound healing.

## 1. Introduction

Wound healing is a fundamental physiological process that naturally restores tissue integrity following structural damage, particularly to the skin [1]. This intricate process involves a series of well-coordinated interactions among various cell types that is mediated by cytokines, chemokines, growth factors, and metabolites, with each playing a critical role in ensuring efficient repair [2]. Traditionally, wound healing is categorized into four overlapping phases [3]. The hemostasis phase, which begins immediately after injury to stop bleeding through clot formation, is followed by the inflammation phase, where immune cells migrate to the wound site to remove pathogens and debris, creating an environment conducive to repair [3]. The proliferation phase involves re-epithelialization, during which new tissue and blood vessels replace damaged structures [3]. Finally, the remodeling phase, or scar maturation, strengthens and reorganizes the newly formed tissue to restore its function and appearance [3]. These phases highlight the complex yet highly coordinated nature of wound healing.

Chronic wounds, such as diabetic wounds, are associated with several abnormalities, including persistent inflammation, impaired neovascularization, reduced angiogenesis, lower levels of growth factors, and hindered collagen synthesis, all of which are linked to cellular oxidative stress caused by hyperglycemia [4]. High glucose levels in the blood, characteristic of diabetes, lead to an overproduction of reactive oxygen species (ROS) and a decrease in antioxidant defenses, which further exacerbates oxidative damage in tissues [5]. This oxidative stress impairs normal wound-healing mechanisms, contributing to delayed wound closure [6]. If these wounds do not heal properly, they can develop into ulcers, such as diabetic foot ulcers, which affect approximately 20% of the diabetic population and can lead to amputation or even death, making it one of the most common complications of diabetes [7]. The slow healing process, recurrent cycles of reinfection, and severe complications associated with this condition have made it a significant public health challenge [4].

Numerous treatments have been employed to address non-healing diabetic wounds, ranging from advanced therapies such as hyperbaric oxygen therapy to innovative wound dressings [8]. The complexity and variability of diabetic wounds necessitate personalized treatment strategies to achieve optimal outcomes. Fespixon^®^ cream (ON101), which contains 1.25% extracts of *Plectranthus amboinicus* (PA-F4; 0.25%) and *Centella asiatica* (S1; 1%) have demonstrated efficacy in treating diabetic foot ulcers by regulating collagen synthesis, promoting the regeneration of damaged cells, and enhancing the proliferation of human keratinocytes [9]. Reduced levels of inflammatory M1 macrophages and increased numbers of reparative M2 macrophages are also related to the effect of Fespixon^®^ cream on the promotion of complete wound healing [10]. The success of Fespixon^®^ cream underscores a broader shift in wound care research, emphasizing the exploration of natural compounds and plant-based therapies.

Hispolon (6-(3,4-dihydroxyphenyl)-4-hydroxyhexa-3,5-dien-2-one; C_12_H_12_O_4_; Figure 1) is a naturally occurring yellow polyphenol with a molecular weight of 220.22 g/mol. It was first extracted in 1996 from the medicinal basidiomycete *Inonotus hispidus* and is recognized for its anti-cancer and immunomodulatory effects [11]. The isolation process involves drying and grinding the mushroom fruiting bodies, followed by organic solvent extraction to produce a crude extract, which is then purified using liquid–liquid extraction or column chromatography. Identification relies on its unique physicochemical characteristics, which are confirmed via spectroscopic techniques [11]. Hispolon has also been extracted from fungi of the *Phellinus* genus, with *P. linteus* yielding the highest concentration—0.1629 mg/g of dried mushroom powder—following ethanol extraction for 6 h [12]. *P. linteus*, a functional food known for its immune-enhancing properties, has been studied clinically. In a trial with participants experiencing reduced immunity, oral administration of *P. linteus* extract at doses of 1000 or 2000 mg/day for eight weeks showed no adverse effects, with improved immune function noted at the 1000 mg/day dosage [13]. The phenolic groups in hispolon’s structure contribute to its strong antioxidative activity [14], underpinning its diverse pharmacological effects, including anti-tumor, antiviral, hepatoprotective, and immunomodulatory properties [15]. Hispolon has demonstrated potent inhibition of both α-glucosidase and aldose reductase in vitro, suggesting its potential as a lead compound in antidiabetic drug development [15]. Additionally, hispolon protects pheochromocytoma cells from oxidative damage by activating nuclear factor erythroid 2-related factor 2 (Nrf-2), a critical transcription factor that guards against oxidative and electrophilic stress [16]. It also reduces inflammation and prevents apoptosis in macrophages exposed to bacterial components such as lipopolysaccharide, lipoteichoic acid, and peptidoglycan by modulating the nuclear factor kappa B signaling pathway, highlighting its potential as a potent anti-inflammatory agent [17]. Although inflammation is essential for initiating wound repair, excessive or prolonged inflammation can impede recovery and increase the risk of chronic non-healing wounds. Therefore, controlling inflammation or shortening its duration can facilitate better wound-healing outcomes [18]. Despite hispolon’s notable anti-inflammatory and antidiabetic properties, its specific role in skin wound healing, particularly in diabetic conditions, remains underexplored and warrants further investigation.

This research aims to comprehensively evaluate the wound-healing properties of hispolon, focusing particularly on its potential application for diabetic wounds. Through a combination of in vitro and in vivo studies, this study seeks to uncover the mechanisms and therapeutic benefits of hispolon in enhancing wound healing under hyperglycemic conditions. In the in vitro phase, L929 skin fibroblast cells will be incubated in a high-glucose environment to simulate the conditions typically observed in diabetic patients. These experiments will assess crucial cellular responses, such as proliferation and migration, under the influence of hispolon, providing insight into its role in cellular repair and regeneration. To translate these findings into a physiological context, in vivo studies will be conducted using streptozotocin (STZ)-induced diabetic rats, a widely recognized model for investigating diabetic complications due to its ability to mimic human hyperglycemia [19]. Three distinct wound models—excision wounds, dead-space wounds, and linear incision wounds—will be employed to evaluate specific parameters of wound healing, including the rate of wound closure, tissue regeneration quality, and tensile strength of newly formed tissues [20]. Additionally, by comparing these results with the established efficacy of Fespixon^®^ cream, this research aims to provide a benchmark for hispolon’s therapeutic potential. The study aspires to not only enhance our understanding of hispolon’s role in wound healing but also to lay the groundwork for its potential application as a novel treatment for diabetic wounds, a condition that poses significant public health challenges worldwide.

## 2. Materials and Methods

### 2.1. Chemicals

Hispolon (Cat. #sc-221726; purity: 98%) was obtained from Santa Cruz Biotechnology, Inc. (Santa Cruz, CA, USA). RPMI 1640 medium (Cat. #R0883), dimethyl sulfoxide (DMSO; Cat. #D8418), 2′,7′-dichlorofluorescin diacetate (DCFH-DA; Cat. #287810), and streptozotocin (STZ; Cat. #S0130) were purchased from Sigma-Aldrich (St. Louis, MO, USA). The XTT Cell Viability Kit (Cat. #9095) was supplied by Cell Signaling Technology Inc. (Danvers, MA, USA). Kits for the lipid peroxidation (LPO) assay (Cat. #ab118970), 8-hydroxy-2′-deoxyguanosine (8-OHdG) ELISA (Cat. #ab201734), BCA protein assay (Cat. #ab102536), tumor necrosis factor (TNF)-α (Cat. #ab236712), interleukin (IL)-6 (Cat. #ab234570), IL-1β (Cat. #ab255730), IL-10 (Cat. #ab214566), fibroblast growth factor (FGF)21 (Cat. #ab223589), transforming growth factor (TGF)-β1 (Cat. #ab119557), epidermal growth factor (EGF; Cat. #ab234560), and vascular endothelial growth factor (VEGF; Cat. #ab209882) were obtained from Abcam plc. (Cambridge, MA, USA). The reduced and oxidized glutathione (GSH/GSSG) assay kit (Cat. #HY-K0311) was procured from MedChemExpress LLC (Monmouth Junction, NJ, USA). Fespixon^®^ cream was supplied by Oneness Biotech Co., Ltd. (Taipei, Taiwan). CD86 antibody (Cat. #188-10036-IHC) and CD206 antibody (Cat. #DS-MB-03769) were purchased from RayBiotech Life, Inc. (Norcross, GA, USA). VECTASTAIN^®^ ABC reagent (Cat. #PK-6100) and 3,3′-diaminobenzidine (Cat. #SK-4100) were sourced from Vector Laboratories (Burlingame, CA, USA). ELISA kits for type I collagen (Cat. #CK-bio-14411) and type III collagen (Cat. #CK-bio-14413) were obtained from Shanghai Coon Koon Biotech Co., Ltd. (Shanghai, China).

### 2.2. Cell Line Culture

L929 mouse fibroblast cells (ATCC # CCL-1) were obtained from the American Type Culture Collection. These cells, stored in the vapor phase of a liquid nitrogen tank, were quickly thawed in a 37 °C water bath. They were then diluted with RPMI 1640 medium, which was supplemented with 10% heat-inactivated fetal bovine serum (FBS), 100 units/mL of penicillin, and 100 μg/mL of streptomycin. The cells were maintained at 37 °C in an incubator with 5% CO_2_. The cell cultures were incubated at 37 °C in a humidified atmosphere containing 5% CO_2_ and 95% air. Passaging of cells occurred every three days, and they were utilized for experiments once they reached approximately 85% confluency.

### 2.3. High-Glucose Stimulation and Treatments

Cells were plated into 6-well plates at a density of 2 × 10^6^ cells per well. Once the cells reached confluence, they were subcultured by detaching them with 0.05% (*w*/*v*) trypsin in phosphate-buffered saline (PBS) at pH 7.4. Prior to the experiments, the cells were incubated in fresh medium containing 1% FBS for 2 h. Glucose is typically added to cell culture media at concentrations ranging from 1 g/L (5.5 mmol/L) to 10 g/L (55 mmol/L). Supplementation with approximately 5.5 mmol/L of d-glucose simulates normal blood glucose levels, while concentrations nearing 10 mmol/L mimic pre-diabetic conditions, and levels exceeding 10 mmol/L reflect diabetic states [21]. For high-glucose functional studies, glucose concentrations in the culture medium were adjusted to 5.5, 11, 22, or 33 mmol/L for durations ranging from 24 to 48 h. In the treatment groups, cells were pretreated with different concentrations of hispolon (2.5, 5, 7.5, or 10 μmol/L) for 1 h. Following pretreatment, the cells were further incubated for 48 h in media containing either normal glucose (5.5 mmol/L) or high glucose (33 mmol/L). The selected hispolon concentration range was based on a previous study showing that these levels of hispolon could inhibit neuronal ferroptosis by enhancing Nrf-2 protein levels, thereby boosting the antioxidative capacity [15]. A stock solution of hispolon (100 μmol/L) was prepared by dissolving the compound in DMSO, and this solution was diluted with the culture medium to achieve the desired concentrations for the experiments. For the control, an equivalent amount of DMSO alone was added, ensuring that the final DMSO concentration in all experimental conditions did not exceed 0.1% (*v*/*v*), which is compatible with the cell lines [22]. After treatment, various parameters were assessed, including cell viability, ROS production, oxidative damage markers such as LPO and 8-OHdG levels, glutathione content, and results from scratch migration assays. Each condition was tested in triplicate, and all experiments were conducted independently at least five times.

### 2.4. Cell Viability Assay

Cell viability was assessed using the XTT Cell Viability Kit according to the manufacturer’s instructions. Cells were plated into 96-well plates at a density of 2 × 10^4^ cells/mL and cultured for 24 h under standard conditions (37 °C, 5% CO_2_) to establish a confluent monolayer. Following this, cells were treated with varying concentrations of hispolon or a vehicle control for 1 h. They were then incubated in a medium containing either normal glucose (5.5 mmol/L) or high glucose (33 mmol/L) for 48 h. At the conclusion of the incubation, 100 μL of fresh medium and 50 μL of XTT/Phenazine Methosulfate (PMS) solution were added to each well, and the plates were incubated for an additional 4 h under the same conditions. Finally, 100 μL of the resulting solution was transferred to a separate plate, and the absorbance at 450 nm was measured using a microplate reader.

### 2.5. Quantification of ROS Production

The cells were plated into 24-well plates, and following the treatment period, they were collected and incubated with 10 μmol/L of the oxidation-sensitive dye DCFH-DA in a serum-free medium at 37 °C for 30 min. Afterward, the cells were washed three times with PBS, and the fluorescence intensity was measured using a multi-detection microplate reader (SpectraMax M5; Molecular Devices, Sunnyvale, CA, USA) with excitation and emission wavelengths of 488 nm and 525 nm, respectively [23].

### 2.6. Measurement of Oxidative Damage to Lipids and DNA

LPO levels were evaluated through the detection of thiobarbituric acid-reactive substances (TBARS) in cell homogenates, following the methodology described in [24]. An LPO assay kit was employed for this purpose, with absorbance readings taken at 532 nm. Additionally, the concentration of 8-OHdG, a recognized marker for oxidative DNA damage [25], was determined using an ELISA kit. All the experimental steps adhered strictly to the manufacturer’s guidelines, with the absorbance measured at 450 nm.

### 2.7. Assessment of the Glutathione Redox State


The GSH/GSSG ratio was measured using a commercial assay kit designed for GSH/GSSG quantification through a spectrophotometric recycling method. The method involves detecting cellular GSH and GSSG levels by monitoring the rate at which the chromogenic product 5-thio-2-nitrobenzoic acid is formed from the substrate 5,5′-dithio-bis(2-nitrobenzoic acid). The resulting absorbance is measured at 412 nm, with the rate of chromophore formation correlating directly with GSH levels.

### 2.8. Scratch Migration Assays

Fibroblast migration was evaluated using the scratch assay in accordance with standard methods [26]. L929 cells were seeded into 6-well plates at a density of 2 × 10^5^ cells/mL and cultured in complete medium until they reached 80–90% confluency. Once the desired confluency was achieved, the cells were incubated at 37 °C with 5% CO_2_ for 24 h. After a 1 h pretreatment with different concentrations of hispolon or a control vehicle, the cells were exposed to media containing either normal glucose (5.5 mmol/L) or high glucose (33 mmol/L) for 48 h. A uniform scratch was then created across the cell monolayer using a sterile 200 μL pipette tip, and cellular debris was removed by washing with PBS. Microscopic images of cell migration were captured at 0 and 24 h post-scratch using an Olympus IX70 microscope (Tokyo, Japan). ImageJ software version 1.38 (NIH, Bethesda, MD, USA) was used to analyze the data.

### 2.9. Animal Model Construction

Male Wistar rats, aged 8 to 10 weeks and weighing 200–250 g, were obtained from the National Laboratory Animal Center in Taipei, Taiwan. They were housed in a controlled environment with a temperature of 25 ± 1 °C and maintained on a 12 h light/dark cycle (lights on at 06:00). The animals had free access to food and water throughout the study. Diabetes was induced by administering 60 mg/kg of STZ via intravenous injection. After three days, blood samples were collected from the tail vein, and plasma glucose levels were measured using an ACCU-CHEK^®^ Active glucometer (Roche Diagnostics, Mannheim, Germany). Rats with plasma glucose concentrations of 250 mg/dL or higher were considered diabetic and included in the study. All procedures were conducted in compliance with the guidelines outlined by the Animal Welfare Act and the National Institutes of Health’s Guide for the Care and Use of Laboratory Animals. The protocol received approval from the Institutional Animal Care and Use Committee (IACUC) at Tajen University (approval number: IACUC 113-18; approval date: 10 March 2024). To uphold ethical standards, all experiments adhered to the principles of the 3Rs—replacement, reduction, and refinement—ensuring humane treatment and the promotion of animal welfare.

### 2.10. Ointment Formulation

Preparation of the simple ointment followed the British Pharmacopoeia guidelines, utilizing 170 g of white soft paraffin, 10 g of hard paraffin, 10 g of cetostearyl alcohol, and 10 g of wool fat [27]. To develop treatment ointments with hispolon concentrations of 2.5% (*w*/*w*) and 5% (*w*/*w*), 2.5 g and 5 g of hispolon were blended into 97.5 g and 95 g of the base ointment, respectively.

### 2.11. Acute Dermal Toxicity Test

Ten healthy Wistar rats and ten diabetic rats were individually housed and allowed a 5-day acclimation period to adapt to laboratory conditions before undergoing acute toxicity testing. Prior to the experiment, the dorsal surface (10%) of each rat was shaved under anesthesia 24 h in advance. The rats, both Wistar and diabetic, were randomly divided into two groups: a control group and a test group, with each group comprising five rats. The control group received a single topical application of 2000 mg/kg of a basic ointment, whereas the test group was treated with an ointment containing 5% (*w*/*w*) hispolon at the same dosage. All procedures adhered to OECD Guideline 402 (Acute Dermal Toxicity: Fixed Dose Procedure) [28]. After application, the rats were closely monitored for any signs of skin toxicity, including irritation, redness, itching, swelling, or behavioral changes, during the first hour. Daily observations continued over a 14-day period to detect any potential adverse skin effects.

### 2.12. In Vivo Wound-Healing Activity

To investigate the wound-healing properties of hispolon, three distinct wound models were employed: excision wounds, dead-space wounds, and incisional wounds. Within each model, the rats were categorized into the following experimental groups for comparison: Group I—normal rats receiving no treatment; Group II—normal rats treated with a sample ointment; Group III—STZ-diabetic rats without treatment; Group IV—STZ-diabetic rats treated with a sample ointment; Group V—STZ-diabetic rats treated with an ointment containing 2.5% (*w*/*w*) hispolon; Group VI—STZ-diabetic rats treated with an ointment containing 5.0% (*w*/*w*) hispolon; and Group VII—STZ-diabetic rats treated with Fespixon^®^ cream. Fespixon^®^ cream includes 1.25% of active compounds derived from PA-F4 (from *P. amboinicus*) and S1 (from *C. asiatica*) combined in a 1:4 ratio and formulated into 15 g of cream per tube.

#### 2.12.1. Excision-Wound Model

An excision-wound model was employed using a well-established protocol [20]. The experiment included seven groups, each comprising eight animals anesthetized via an open-mask technique with ether as the anesthetic agent. The dorsal area of the rats was shaved, and a single excision wound, designated as day “0”, was created by removing a 500 mm^2^ section of full-thickness skin from a predefined location. The wounds were left exposed to an open environment. Test materials, including the simple ointment, hispolon ointments at 2.5% and 5% (*w*/*w*) concentrations, and Fespixon^®^ cream, were applied topically at a dose of approximately 0.2 g per wound daily until complete healing occurred. Wound areas were assessed every two days by tracing onto graph paper marked with millimeter scales. Wound contraction was calculated as the percentage reduction in the wound surface area over time. The duration required for epithelialization was determined by recording the number of days until the eschar fully detached, leaving no open wound. Skin samples from each group in the excision-wound model were collected on days 7 and 14 post-injury for immunohistochemical and histopathological analyses, respectively.

#### 2.12.2. Dead-Space-Wound Model

The rats were put under anesthesia, and a 1 cm incision was made on the dorsolumbar region of their backs. Two polypropylene tubes, each sized 0.5 × 2.5 cm^2^, were inserted into the dead space on both sides of the lumbar region. The incisions were then closed with sutures [20]. The wounds received topical treatments once daily with 0.2 g of either a basic ointment, hispolon ointments at 2.5% and 5% (*w*/*w*) concentrations, or Fespixon^®^ cream. On the seventh day post-injury, the animals were sacrificed, and the granulation tissue formed around the implanted tubes was carefully excised. This tissue was subsequently analyzed to evaluate the levels of pro-inflammatory and anti-inflammatory cytokines, measure the content of growth factors, and assess the properties of collagen.

#### 2.12.3. Incision-Wound Model

An incisional skin-wound model was established following standard methodology [20]. In summary, after anesthesia administration, two parallel incisions, each measuring 6 cm in length, were made through the full-thickness skin, positioned 1 cm from the midline on either side of the vertebral column. The incision edges were closed using Acos™ disposable stainless steel skin staplers (Sunmedix Co., Ltd., Namyangjusi, Gyeonggido, Republic of Korea) spaced 1 cm apart. Topical treatments were applied daily for 14 days, using 0.2 g of either a simple ointment, hispolon ointment at 2.5% or 5% (*w*/*w*), or Fespixon^®^ cream. On the 10th day, the skin staplers were removed, but the topical treatments continued. On the 14th day, the animals were euthanized under anesthesia, and the tensile strength of the wounds was evaluated using a tensiometer. The device measured the force (in grams) required to rupture a defined skin area (mm^2^). The wound-breaking strength (WBS) was calculated as the force in grams per square millimeter (g/mm^2^) necessary to break the wound area.

### 2.13. Immunohistochemistry Analysis

On the 7th day post-injury, skin samples from the healed wound area of an excision-wound rat model were collected, fixed in formalin, and embedded in paraffin. These samples were then analyzed for the presence of macrophages through immunohistochemical staining. M1 macrophages were identified by CD86 expression, while M2 macrophages were identified by CD206 expression. After deparaffinization and hydration, the slides were rinsed with Tris-buffered saline (TBS; 10 mmol/L Tris HCl, 0.85% NaCl, pH 7.2). To inhibit endogenous peroxidase activity, the samples were treated with a solution containing methanol and 0.3% hydrogen peroxide in methanol. Subsequently, the slides were incubated overnight at 4 °C with either the CD86 antibody or the CD206 antibody. After incubation, the slides were carefully washed with TBS to remove any residual reagents. A secondary antibody conjugated with horseradish peroxidase was utilized, and the slides were incubated for 1 h at room temperature. Subsequently, the slides were treated with the VECTASTAIN^®^ ABC reagent, followed by color development using 3,3′-diaminobenzidine. Immunohistochemical evaluations were conducted independently by two investigators in a double-blinded manner. The stained regions were analyzed in pixels using ImageJ 1.38 software (NIH, Bethesda, MD, USA). The positive staining observed in the treated group was compared to the staining levels observed in the untreated control group of normal rats.

### 2.14. Histopathological Studies

Skin samples were fixed in a 10% neutral-buffered formalin solution, which was replaced every two days until the tissues became firm. The specimens were subsequently embedded in paraffin, and 3 μm sections were cut and stained with hematoxylin and eosin (H&E) to assess their general morphology. A Carl Zeiss Axio Imager M2 light microscope was used to examine various features, such as collagen formation, fibroblast activity, angiogenesis, and granulation tissue development. These parameters were assessed based on their intensity and categorized into four grades: 1 (minimal; 1–25%), 2 (mild; 26–50%), 3 (moderate; 51–75%), and 4 (high; 76–100%) [29].

### 2.15. Assessment of Inflammatory Markers and Growth Factors

On the seventh day following the establishment of the dead-space-wound model, granulation tissue surrounding the implanted tubes was harvested. This tissue was homogenized in a 0.15 mol/L KCl solution and subjected to centrifugation at 8000 rpm for 10 min. The cell-free supernatant obtained after centrifugation was used for subsequent analyses. Levels of cytokines such as TNF-α, IL-6, IL-1β, and IL-10 were assessed using commercially available ELISA kits. Concentrations of FGF21, TGF-β1, EGF, and VEGF were also determined using ELISA kits from the same supplier. All procedures adhered strictly to the manufacturer’s protocols, and the absorbance was measured at 450 nm using a SpectraMax M5 microplate reader (Molecular Devices, Sunnyvale, CA, USA). The total protein content of the samples was determined using a BCA protein assay kit. Protein concentrations were calculated based on a colorimetric change from green to purple, which correlates with changes in protein levels, and the absorbance was measured at 562 nm. Results were expressed as picograms of protein per milligram (pg/mg) of sample. To ensure data reliability, all experiments were conducted in triplicate.

### 2.16. Quantitative Estimation of the Collagen Content

Granulation tissue surrounding the implanted tubes was collected on day 7 post-wounding in the dead-space-wound model to evaluate the collagen content. For each sample, 40 mg of dried granulation tissue was placed in a tube with 1 mL of 6N HCl and hydrolyzed in a boiling water bath for 12 h daily over two consecutive days. Following hydrolysis, the solution was cooled, and excess acid was neutralized using 10N NaOH with phenolphthalein as the pH indicator. The neutralized solution was then diluted with distilled water to a final concentration of 20 mg/mL. Type I and type III collagen levels were quantified using commercial ELISA kits, with the absorbance measured at 450 nm via a SpectraMax M5 microplate reader (Molecular Devices, Sunnyvale, CA, USA). The results were expressed as nanograms of collagen per milligram of protein. The total protein content of the samples was determined using a BCA protein assay kit.

### 2.17. Statistical Analysis

Statistical analyses were performed using SigmaPlot Version 14.0 (Systat Software Inc., San Jose, CA, USA). Data are reported as the mean ± standard deviation (SD). For comparisons between two sample groups, the Student’s *t*-test was applied, while one-way ANOVA was used for comparisons across multiple groups. A *p*-value of <0.05 was regarded as statistically significant.

## 3. Results

### 3.1. Effect of Hispolon on the Viability of Fibroblasts Exposed to High Glucose Concentrations

#### 3.1.1. Hispolon Promotes Fibroblast Viability

The effects of increasing glucose concentrations on fibroblast viability were assessed over time. After 24 h, glucose levels of 11, 22, and 33 mmol/L showed no significant impact on cell viability compared with the control group treated with 5.5 mmol/L glucose (Figure 2A). In contrast, after 48 h, treatments with 22 and 33 mmol/L glucose resulted in decreased cell viability compared with the control (5.5 mmol/L glucose; Figure 2A). Notably, exposure to 33 mmol/L glucose for 48 h led to an approximately 49% reduction in cell viability (Figure 2A), making this condition ideal for subsequent experiments aimed at investigating more pronounced effects.

Hispolon treatment at concentrations of 2.5, 5, 7.5, or 10 μmol/L for 48 h did not affect the number of L929 cells under normal-glucose conditions, maintaining their cell viability at above 90% (Figure 2B). Under high-glucose conditions (33 mmol/L), L929 cell viability increased progressively with higher hispolon concentrations, achieving a maximum survival rate of 91.2% at 10 μmol/L (Figure 2B).

#### 3.1.2. Hispolon Reduces Oxidative Damage and Enhances Antioxidative Defense

The intracellular levels of LPO and 8-OHdG were increased by 3.2-fold and 2.8-fold, respectively, in cells exposed to high glucose compared with those in the normal-glucose control group (Figure 3B,C). Hispolon pretreatment (10 μmol/L) resulted in a 55.6% reduction in LPO levels and a 48.3% decrease in 8-OHdG levels when compared with the vehicle-treated group (Figure 3B,C). Additionally, high-glucose exposure caused a reduction in the GSH/GSSG ratio, which was restored in a dose-dependent manner following hispolon pretreatment (Figure 3D).

### 3.2. Effects of Hispolon on the Scratch Wound Closure in Fibroblast Exposed to High Glucose Concentrations

Figure 4A presents representative live-cell micrographs of L929 cells after 24 h of culture under normal-glucose conditions (5.5 mmol/L) or high-glucose conditions (33 mmol/L). The quantified migration distances of these cells are shown in Figure 4B. Exposure to high glucose significantly decreased cell migration to 52.2 ± 4.9% compared with the normal-glucose control group (Figure 4B). Treatment with hispolon improved cell migration under high-glucose conditions in a dose-dependent manner (2.5, 5, 7.5, and 10 μmol/L). At the highest concentration (10 μmol/L), hispolon restored migration to 86.3 ± 6.7% of the distance observed in the normal-glucose control group (Figure 4B).

### 3.3. Effects of Hispolon on Wound Healing in Diabetic Rats Using an Excision-Wound Model

#### 3.3.1. Hispolon Facilitates Wound Contraction

Plasma glucose levels in STZ-diabetic rats were significantly elevated compared with those in normal rats (Figure 5A). Topical treatments, including the simple ointment and hispolon ointments at concentrations of 2.5% (*w*/*w*) and 5% (*w*/*w*), did not affect plasma glucose levels in the STZ-diabetic rats at any point during the experimental period (Figure 5A). Likewise, the application of Fespixon^®^ cream had no impact on plasma glucose levels throughout the study duration (Figure 5A).

The epithelialization period was prolonged in untreated STZ-diabetic rats compared with untreated normal rats (Figure 5C). In STZ-diabetic rats, treatment with hispolon ointment at 2.5% (*w*/*w*) and 5% (*w*/*w*) concentrations shortened the epithelialization time to 20.3 and 22.9 days, respectively, in contrast to the 29.6 days observed in the group receiving the simple ointment (Figure 5C). The epithelialization time in STZ-diabetic rats treated with 5% (*w*/*w*) hispolon ointment was comparable to that of the group treated with Fespixon^®^ cream, which showed a duration of 19.4 days.

#### 3.3.2. Hispolon Regulates Macrophage Polarization

As shown in Figure 6A, seven days after wound formation in the excision-wound model, both untreated STZ-diabetic rats and those receiving the simple ointment exhibited a higher prevalence of M1 macrophages (CD86-positive cells) compared with untreated normal rats. The application of 5% (*w*/*w*) hispolon ointment or Fespixon^®^ cream significantly reduced CD86 expression levels. Conversely, on day 7, M2 macrophage (CD206-positive cell) expression in excision wounds was lower in STZ-diabetic rats relative to untreated normal rats. However, treatment with 5% (*w*/*w*) hispolon ointment or Fespixon^®^ cream increased CD206 levels (Figure 6A). Use of the simple ointment did not alter M1 or M2 macrophage levels in either the normal or STZ-diabetic rats (Figure 6A).

#### 3.3.3. Hispolon Promotes Tissue Regeneration

Figure 6B depicts the histopathological features of wound healing 14 days after injury. In untreated STZ-diabetic rats, the fibroblasts appeared disorganized, collagen fiber deposition was reduced, and angiogenesis was impaired compared with untreated normal rats. Application of Fespixon^®^ cream in STZ-diabetic rats promoted tissue regeneration, enhanced collagen synthesis, and increased granulation tissue development. Wounds in STZ-diabetic rats that were treated with hispolon ointment exhibited greater fibroblast activity and improved capillary formation, with the 5% (*w*/*w*) concentration proving more effective than the 2.5% (*w*/*w*) concentration. The simple ointment had no observable impact on fibroblast proliferation, collagen deposition, or angiogenesis in either normal or STZ-diabetic rats. Detailed histopathological assessments and scoring for the experimental groups are shown in the lower panel of Figure 6B.

### 3.4. Effects of Hispolon on Granulation Tissue Formation in Diabetic Rats with a Dead-Space-Wound Model

#### 3.4.1. Hispolon Ameliorates Inflammatory Responses

After 7 days, the levels of TNF-α (125.5 ± 13.2 pg/mg protein vs. 123.3 ± 12.1 pg/mg protein), IL-6 (78.4 ± 8.1 pg/mg protein vs. 75.3 ± 7.2 pg/mg protein), IL-1β (50.3 ± 6.9 pg/mg protein vs. 58.2 ± 7.1 pg/mg protein), and IL-10 (133.2 ± 10.9 pg/mg protein vs. 134.3 ± 9.4 pg/mg protein) in granulation tissue from dead-space wounds remained consistent between untreated normal rats and those treated with the simple ointment (Figure 7A). In contrast, untreated STZ-diabetic rats and those receiving the simple ointment exhibited higher concentrations of TNF-α (256.8 ± 11.7 pg/mg protein vs. 259.4 ± 14.2 pg/mg protein), IL-6 (138.2 ± 13.9 pg/mg protein vs. 140.6 ± 14.7 pg/mg protein), and IL-1β (100.3 ± 6.8 pg/mg protein vs. 106.3 ± 7.1 pg/mg protein) alongside reduced IL-10 levels (76.2 ± 8.7 pg/mg protein vs. 74.2 ± 7.6 pg/mg protein) compared with untreated normal rats after the same period (Figure 7A).

When STZ-diabetic rats were treated with 5% (*w*/*w*) hispolon ointment, TNF-α (172.3 ± 16.5 pg/mg protein), IL-6 (101.9 ± 9.8 pg/mg protein), and IL-1β (84.4 ± 8.2 pg/mg protein) levels declined, while IL-10 levels increased (103.6 ± 10.4 pg/mg protein). These changes closely resembled the effects observed in the Fespixon^®^ cream-treated group (Figure 7A). In the latter group, TNF-α levels were 165.3 ± 14.6 pg/mg protein, IL-6 levels were 95.9 ± 8.7 pg/mg protein, IL-1β levels were 70.3 ± 9.3 pg/mg protein, and IL-10 levels were 119.7 ± 13.9 pg/mg protein (Figure 7B).

#### 3.4.2. Hispolon Promotes an Increased Concentration of Growth Factors

On day 7 post-wounding, there was no significant variation in the levels of FGF21 (207.7 ± 14.2 pg/mg protein vs. 201.4 ± 15.3 pg/mg protein), TGF-β1 (178.3 ± 14.9 pg/mg protein vs. 175.7 ± 16.3 pg/mg protein), EGF (180.3 ± 16.7 pg/mg protein vs. 185.2 ± 17.3 pg/mg protein), and VEGF (233.6 ± 15.4 pg/mg protein vs. 234.3 ± 16.5 pg/mg protein) in the granulation tissue between normal rats treated with the simple ointment and untreated normal rats (Figure 7B). In the dead-space-wound model, both untreated STZ-diabetic rats and those treated with the simple ointment exhibited reduced FGF21 levels (56.8 ± 5.2 pg/mg protein vs. 59.4 ± 7.1 pg/mg protein) and lower concentrations of TGF-β1 (35.3 ± 4.3 pg/mg protein vs. 37.4 ± 5.4 pg/mg protein), EGF (60.3 ± 7.1 pg/mg protein vs. 62.3 ± 6.8 pg/mg protein), and VEGF (78.2 ± 8.7 pg/mg protein vs. 75.2 ± 9.3 pg/mg protein) in granulation tissue compared with untreated normal rats (Figure 7B). Treatment with 5% (*w*/*w*) hispolon ointment in STZ-diabetic rats enhanced FGF21 levels (156.4 ± 15.1 pg/mg protein), as well as the levels of TGF-β1 (101.9 ± 10.5 pg/mg protein), EGF (120.4 ± 12.3 pg/mg protein), and VEGF (153.6 ± 14.4 pg/mg protein), reaching levels comparable to those achieved with Fespixon^®^ cream. Rats treated with Fespixon^®^ cream demonstrated FGF21 levels of 166.5 ± 15.9 pg/mg protein, TGF-β1 levels of 141.1 ± 14.7 pg/mg protein, EGF levels of 151.2 ± 17.3 pg/mg protein, and VEGF levels of 179.7 ± 15.8 pg/mg protein (Figure 7B).

#### 3.4.3. Hispolon Enhances Collagen Production

By the 7th day, the granulation tissue of STZ-diabetic rats with dead-space wounds exhibited reduced levels of type I collagen (76.7 ± 7.1 μg/mg protein) and type III collagen (44.1 ± 5.6 μg/mg protein) compared with normal rats, which had type I collagen levels of 150.3 ± 8.2 μg/mg protein and type III collagen levels of 95.6 ± 8.2 μg/mg protein (Figure 7C). Treatment with 5% (*w*/*w*) hispolon ointment or Fespixon^®^ cream increased type I collagen levels to 103.3 ± 10.1 μg/mg protein and 119.3 ± 10.1 μg/mg protein, respectively, and type III collagen levels to 68.3 ± 7.1 μg/mg protein and 83.4 ± 6.4 μg/mg protein, respectively, compared with the simple ointment group, which showed 75.5 ± 6.8 μg/mg protein of type I collagen and 46.1 ± 5.4 μg/mg protein of type III collagen (Figure 7C). In normal rats, there was no significant variation in the levels of Type I or Type III collagen in granulation tissue after 7 days, regardless of whether the simple ointment was applied or not (Figure 7C).

### 3.5. Effects on Wound-Breaking Strength (WBS) in Diabetic Rats with an Incisional-Wound Model

By the 14th day post-injury, the WBS in untreated STZ-diabetic rats (202.1 ± 10.6 g/mm^2^) was lower than that of untreated normal rats (401.2 ± 14.7 g/mm^2^). Application of the simple ointment did not improve the WBS in STZ-diabetic rats (209.3 ± 12.3 g/mm^2^; Figure 8). Similarly, there was no difference in the WBS between normal rats treated with the simple ointment and untreated normal rats (398.6 ± 15.3 g/mm^2^ vs. 401.2 ± 14.7 g/mm^2^, respectively; Figure 8). In STZ-diabetic rats, treatment with hispolon ointment at concentrations of 2.5% (*w*/*w*) and 5% (*w*/*w*) resulted in a 1.3-fold (263.2 ± 10.2 g/mm^2^) and 1.6-fold (315.2 ± 11.5 g/mm^2^) increase in the WBS, respectively, compared with untreated STZ-diabetic rats. For comparison, treatment with Fespixon^®^ cream led to a 1.7-fold increase in the WBS (337.1 ± 12.6 g/mm^2^; Figure 8).

### 3.6. In Vivo Acute Dermal Toxicity Studies

In an acute toxicity assessment, normal and STZ-diabetic rats were treated with either a simple ointment or a 5% (*w*/*w*) hispolon-based ointment at a maximum dosage of 2000 mg/kg. Observations revealed no abnormalities in the fur, eyes, or mucous membranes and no adverse skin reactions, such as redness, swelling, irritation, itching, or behavioral disturbances, were noted in either group. Additionally, no mortalities occurred during the duration of the study.

## 4. Discussion

Fibroblasts play a critical role in wound healing by aiding re-epithelialization, driving wound closure, and promoting their own proliferation and migration [30]. Our findings showed that exposure to high-glucose conditions significantly increased ROS levels and caused parallel lipid and DNA oxidation in fibroblast L929 cells. The intracellular ratio of GSH to GSSG is a well-established indicator of cellular oxidative stress [31]. In our study, high-glucose exposure led to a marked decrease in the GSH/GSSG ratio in L929 cells, reflecting heightened oxidative stress and demonstrating cytotoxic effects by reducing cell viability. This model not only sheds light on the mechanisms underlying hyperglycemia-related wound-healing impairments but also provides a valuable tool for evaluating therapeutic approaches aimed at mitigating oxidative stress and improving wound repair outcomes.

Studies have demonstrated that hispolon supports fibroblast viability under high-glucose conditions, potentially by regulating the interplay between oxidative stress and the cellular antioxidative defense mechanisms. Elevated glucose levels are known to hinder cell migration, likely due to increased oxidative stress, which could explain the delayed wound healing often observed in diabetic patients [32]. Our findings reveal that hispolon mitigates oxidative stress induced by high glucose, thereby promoting cell migration, as evidenced by the results of a wound scratch assay performed on L929 cells. These results indicate that hispolon not only preserves fibroblast viability in high-glucose environments but also facilitates their proliferation and migration, both of which are critical for effective wound healing. Nonetheless, a limitation of this study lies in its exclusive focus on fibroblasts, which may not fully represent the multifaceted nature of the wound-healing process. Future research should incorporate diabetic animal models to investigate the precise mechanisms by which hispolon influences wound repair under more complex physiological conditions.

Excisional wounds, created by surgically removing the epidermis, dermis, and subcutaneous fat layers, are widely used as models to study wound contraction and epithelialization, providing critical insights into the wound-healing process [20]. In this study, excisional-wound models were utilized to investigate the therapeutic potential of hispolon ointment in promoting wound repair under diabetic conditions. STZ-diabetic rats treated with hispolon demonstrated smaller wound sizes and shorter epithelialization periods compared with those in untreated controls, suggesting that hispolon facilitates wound contraction and expedites the healing process. Given the multifaceted nature of wound healing, which involves overlapping phases of inflammation, proliferation, and remodeling, a complementary dead-space granuloma model was employed [33]. This model offers valuable information on the development and maturation of granulation tissue, a key component for successful wound closure. By focusing on the granulation tissue within dead-space wounds, the study provides a comprehensive assessment of hispolon’s impact on inflammatory regulation, cytokine activity, and connective tissue formation.

Inflammatory processes play a crucial role in regulating wound healing. Macrophages, as key inflammatory cells at wound sites, demonstrate remarkable plasticity through a process known as macrophage polarization. This allows them to switch between two distinct phenotypes that either sustain or resolve inflammation during wound repair [34]. M1 macrophages, also referred to as pro-inflammatory macrophages, primarily produce cytokines such as IL-1β, IL-6, and TNF-α, which intensify inflammation and may exacerbate tissue damage [35]. Conversely, M2 macrophages, recognized for their pro-repair functions, secrete anti-inflammatory cytokines like IL-10 as well as growth factors, including TGF-β, EGF, and VEGF, which are essential for cell proliferation, tissue remodeling, and repair [35]. In our study, granulation tissue from STZ-diabetic rats showed elevated levels of pro-inflammatory cytokines (TNF-α, IL-6, IL-1β) alongside reduced levels of the anti-inflammatory cytokine IL-10 and growth factors such as TGF-β, EGF, and VEGF. Treatment with hispolon ointment significantly reduced the expression of pro-inflammatory cytokines while enhancing anti-inflammatory cytokines and growth factors. Immunohistochemical analysis of excision wounds revealed higher expression of the M1 macrophage marker CD86 and lower expression of the M2 macrophage marker CD206 in STZ-diabetic rats compared with normal controls. Hispolon ointment treatment decreased the population of M1 macrophages while increasing the presence of M2 macrophages. These results suggest that hispolon ointment exerts anti-inflammatory effects by modulating macrophage polarization, shifting from the pro-inflammatory M1 phenotype to the anti-inflammatory M2 phenotype, thereby promoting improved wound healing in STZ-diabetic rats.

The transition from the inflammatory phase to the proliferative phase typically occurs between days 4 and 21 [36]. During the proliferative phase, collagen production and deposition play a pivotal role in strengthening and supporting the healing tissue [37]. Initially, collagen III is synthesized, which is gradually replaced by collagen I, the dominant form in the skin [37]. At the onset of granulation tissue formation, collagen fibers are arranged randomly; over time, covalent cross-linking enhances their structural complexity, leading to reorganization, which restores tensile strength [38]. Consequently, stimulating granulation tissue formation and activating collagen synthesis during this phase are critical objectives for developing wound-healing therapies [39]. Our study revealed that by at day 7 post-wounding, the application of hispolon ointment significantly elevated levels of both type I and type III collagen, consistent with the early proliferation phase—a key period in tissue integrity restoration. By day 14, histopathological analysis had corroborated these findings, showing extensive collagen deposition and well-formed granulation tissue in STZ-diabetic rats treated with hispolon. These results indicate enhanced fibroblast activity and suggest that hispolon therapy not only increases collagen content but also improves its structural organization, thereby facilitating tissue repair and accelerating wound closure, particularly in diabetic wounds with delayed healing. Additional evidence from an incisional skin-wound model demonstrated that hispolon ointment improved the wound-breaking strength in diabetic rats by enhancing the collagen architecture. This improvement in wound integrity underscores the potential of hispolon as a therapeutic agent for chronic diabetic wounds.

When developing a topical medication, it is essential to ensure both its efficacy in promoting wound healing and its safety through thorough evaluation [40]. In an acute toxicity study conducted in rats, a high-dose application of 5% hispolon ointment (2000 mg/kg) revealed no adverse effects, systemic toxicity, behavioral abnormalities, or fatalities. These findings highlight the robust dermatological safety profile of hispolon ointment, reinforcing its potential as a promising candidate for topical therapeutic use.

Monitoring plasma glucose levels is a critical tool for understanding and managing diabetes, as persistent hyperglycemia is a hallmark of the disease and indicates its severity and progression [41]. In this study, evaluating hyperglycemia was a primary focus in assessing therapeutic effectiveness in diabetes management and the prevention of complications. Interestingly, despite various treatments, plasma glucose levels in STZ-diabetic rats remained unchanged, suggesting that hispolon facilitates wound healing through mechanisms unrelated to glucose reduction. Hispolon supports the wound-healing process by counteracting glucose-induced oxidative stress, targeting essential pathways necessary for effective tissue repair. It specifically promotes macrophage polarization toward the M2 phenotype, which suppresses inflammatory responses and enhances collagen synthesis. This, in turn, strengthens the tissue structure and accelerates complete wound recovery. The wound-healing benefits of hispolon ointment in diabetic wounds were found to be comparable to those of Fespixon^®^ cream [9,10]. Both ointments and creams are widely utilized in therapeutic and cosmetic treatments, differing mainly in their oil-to-water ratio. Creams, with their higher water content, are absorbed quickly, while ointments, with more oil, form a protective barrier that retains moisture and prolongs skin contact [42]. Hispolon could also be developed into alternative formulations, such as gels for less greasy application or transdermal patches for controlled release. These diverse options address varying therapeutic requirements. Future research should focus on optimizing hispolon formulations by examining its absorption characteristics, hydration properties, barrier effects, efficacy, bioavailability, and potential for anti-diabetic applications.

## 5. Conclusions

This study highlights the potential of hispolon ointment in promoting diabetic wound healing. Hispolon lowers ROS levels, restores the GSH/GSSG balance, and facilitates the shift of macrophages from the pro-inflammatory M1 phenotype to the pro-healing M2 phenotype, thereby increasing the levels of anti-inflammatory cytokines (IL-10) and growth factors (TGF-β, VEGF). Furthermore, hispolon enhances fibroblast migration and collagen synthesis (types I and III), strengthening granulation tissue and accelerating wound closure, particularly in diabetic conditions. These mechanisms establish hispolon as a promising agent for diabetic wound management. Future research should aim to elucidate its molecular mechanisms, optimize formulations for improved delivery and bioavailability, and conduct long-term safety evaluations. Comparative studies with established therapies and explorations of its applications across various wound types and populations may further expand its potential. Ultimately, clinical trials are essential to confirm hispolon’s efficacy and safety, paving the way for its adoption as a natural, effective solution for wound care.

## Figures and Tables

**Figure 1 nutrients-17-00266-f001:**
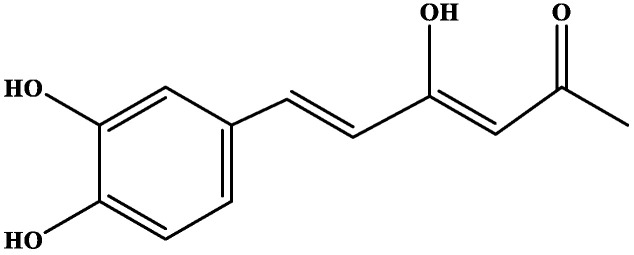
Chemical structure of hispolon.

**Figure 2 nutrients-17-00266-f002:**
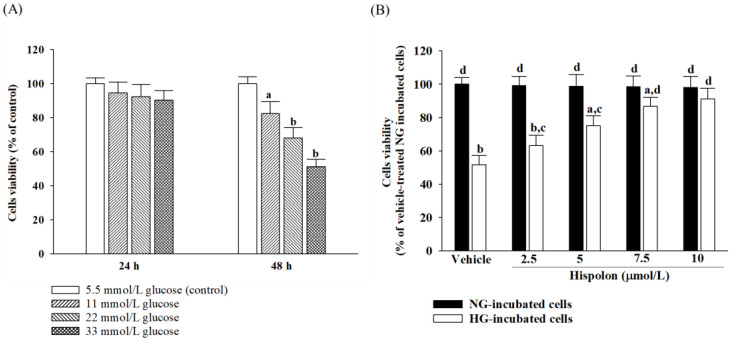
Effect of hispolon on cell viability in L929 cells. (**A**) Cell viability in L929 cells cultured with glucose concentrations ranging from 5.5, 11, and 22 mmol/L to 33 mmol/L for 24 h or 48 h. (**B**) Cells were pretreated with different concentrations of hispolon (2.5, 5, 7.5, or 10 μmol/L) for 1 h and then exposed to normal glucose (5.5 mmol/L; NG) or high glucose (33 mmol/L; HG) for an additional 48 h. Cell viability was determined by the XTT assay. The results are presented as the mean ± SD of five independent experiments (*n* = 5), each of which was performed in triplicate. Statistical significance is indicated as follows: ^a^
*p* < 0.05 and ^b^
*p* < 0.01, compared with the data from vehicle-treated NG-incubated cells (control group); ^c^
*p* < 0.05 and ^d^
*p* < 0.01, compared with the data from vehicle-treated HG-incubated cells.

**Figure 3 nutrients-17-00266-f003:**
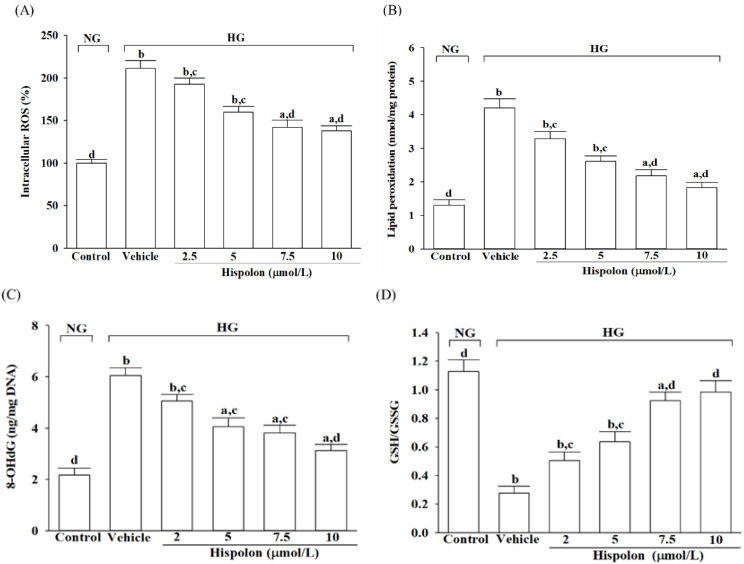
Influence of hispolon on oxidative stress and antioxidative defense responses in L929 cells under high-glucose conditions. L929 cells were pretreated with hispolon at varying concentrations (2.5, 5, 7.5, or 10 μmol/L) for 1 h before being subjected to either normal-glucose (5.5 mmol/L; NG) or high-glucose (33 mmol/L; HG) conditions for an additional 48 h. (**A**) Intracellular ROS levels were assessed using the oxidation-sensitive fluorescent probe DCFH-DA. (**B**) Lipid peroxidation was evaluated via the TBARS assay. (**C**) Levels of 8-OHdG were quantified utilizing a commercially-supplied ELISA kit. (**D**) The ratio of GSH to GSSG in cells was assessed using a commercially provided kit. All experiments were conducted in triplicate, and results are expressed as the mean ± SD from five independent experiments (*n* = 5). Statistical significance is indicated as follows: ^a^
*p* < 0.05 and ^b^
*p* < 0.01, compared with the data from vehicle-treated NG-incubated cells (control group); ^c^
*p* < 0.05 and ^d^
*p* < 0.01, compared with the data from vehicle-treated HG-incubated cells.

**Figure 4 nutrients-17-00266-f004:**
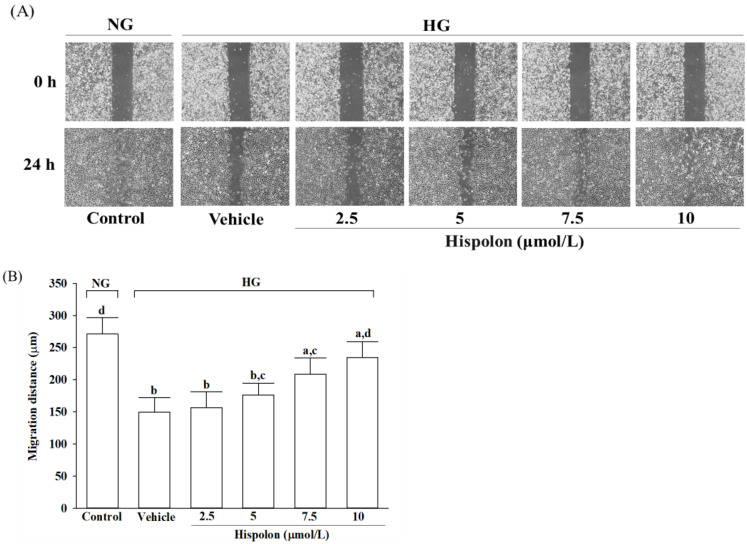
Effect of hispolon on cell migration in high-glucose environments. L929 cells were pretreated with different hispolon concentrations (2.5, 5, 7.5, or 10 μmol/L) for 1 h, followed by exposure to either normal glucose (5.5 mmol/L, NG) or high glucose (33 mmol/L, HG) for 48 h. (**A**) A scratch migration assay was performed, and optical images showing cells migrating into the wound gap were taken at 0 and 24 h at a 100× magnification. (**B**) The migration distance of the cells was measured using ImageJ software version 1.38. All experiments were conducted in triplicate, and the results are expressed as the mean ± SD from five independent experiments (*n* = 5). Statistical significance is indicated as follows: ^a^
*p* < 0.05 and ^b^
*p* < 0.01, compared with the data from vehicle-treated NG-incubated cells (control group); ^c^
*p* < 0.05 and ^d^
*p* < 0.01, compared with the data from vehicle-treated HG-incubated cells.

**Figure 5 nutrients-17-00266-f005:**
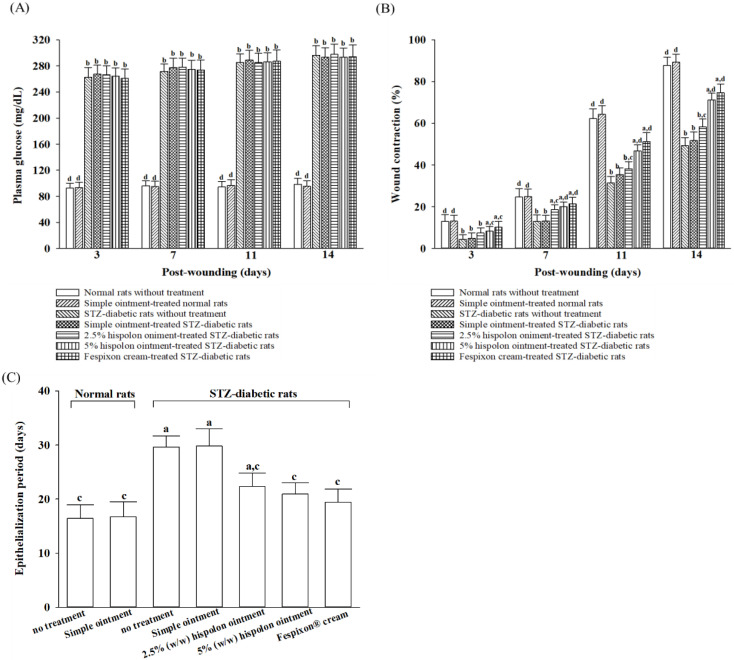
Effects of hispolon on plasma glucose levels and wound contraction in excision wounds in rats. (**A**) Plasma glucose concentration, (**B**) percentage of wound contraction, and (**C**) epithelialization period was measured in STZ-diabetic rats treated with experimental ointments using an excision-wound model. Values (mean ± SD) were obtained from each group of 8 animals. ^a^
*p* < 0.05 and ^b^
*p* < 0.01, compared with the values of normal rats without treatment on the indicated post-wounding day. ^c^
*p* < 0.05 and ^d^
*p* < 0.01, compared with the values of STZ-diabetic rats without treatment on the indicated post-wounding day.

**Figure 6 nutrients-17-00266-f006:**
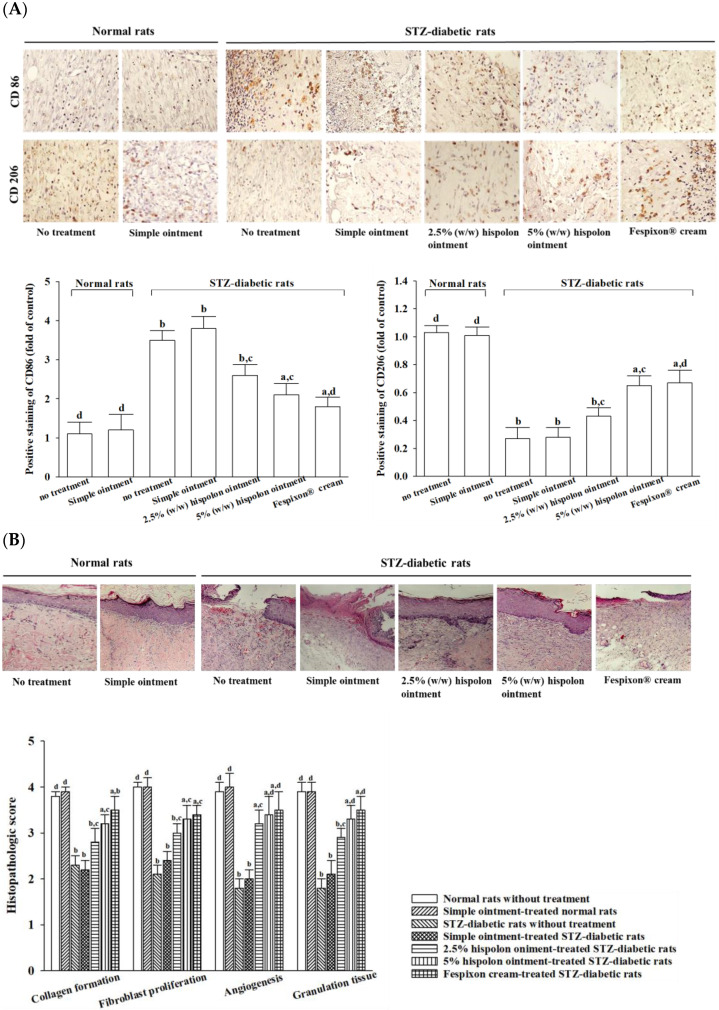
Effects of hispolon on macrophage phenotype regulation and tissue regeneration in rat excision wounds. (**A**) Immunohistochemistry was performed on day 7 post-wounding to identify M1 and M2 macrophage populations in the excision wounds. The upper panel displays immunohistochemical staining images, while the lower panel presents quantified data of the positive staining (dark brown) for CD86 and CD206, indicating M1 and M2 macrophages, respectively. Photomicrographs were captured at ×100 magnification. The positive staining observed in the treated group was compared to the staining values recorded in the untreated control group of normal rats, (**B**) Histological analysis of the wound healing was performed on day 14 post-surgery using hematoxylin and eosin staining. The upper panel presents photomicrographs at ×100 magnification. The lower panel provides the corresponding histopathological scores for the experimental groups. Values (mean ± SD) were obtained from each group of 8 animals. ^a^
*p* < 0.05 and ^b^
*p* < 0.01, compared with the values of normal rats without treatment (control). ^c^
*p* < 0.05 and ^d^
*p* < 0.01, compared with the values of STZ-diabetic rats without treatment.

**Figure 7 nutrients-17-00266-f007:**
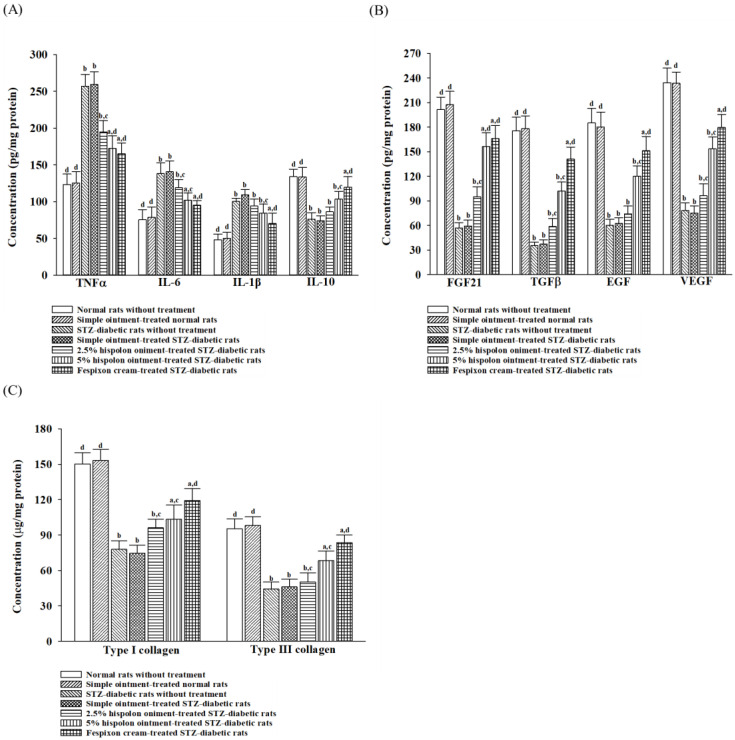
Effects of hispolon on pro-inflammatory and anti-inflammatory cytokine levels, growth factor concentrations, and collagen properties in granulation tissue of dead-space wounds in rats after 7 days. (**A**) The levels of pro-inflammatory cytokines (TNF-α, IL-6, IL-1β) and the anti-inflammatory cytokine IL-10 were quantified. (**B**) The concentrations of growth factors, including FGF21, TGF-β1, EGF, and VEGF, were measured. (**C**) The properties of collagen, focusing on Type I and Type III collagen, were analyzed. Values (mean ± SD) were obtained from each group of 8 animals. ^a^
*p* < 0.05 and ^b^
*p* < 0.01, compared with the values of normal rats without treatment. ^c^
*p* < 0.05 and ^d^
*p* < 0.01, compared with the values of STZ-diabetic rats without treatment.

**Figure 8 nutrients-17-00266-f008:**
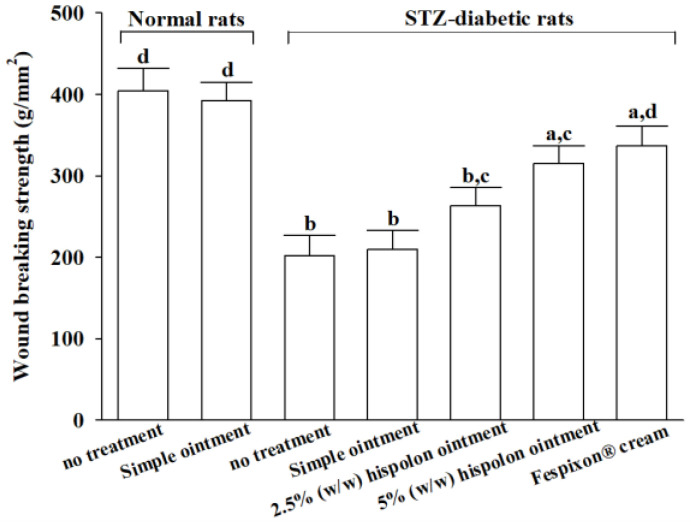
Effects of hispolon on the wound-breaking strength (WBS) in an incision-wound model. Values (mean ± SD) were obtained from each group of 8 animals. ^a^
*p* < 0.05 and ^b^
*p* < 0.01, compared with the values of normal rats without treatment. ^c^
*p* < 0.05 and ^d^
*p* < 0.01, compared to the values of STZ-diabetic rats without treatment.

## Data Availability

The data that support the findings of this study are available from the corresponding author upon reasonable request.

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
