# Peer review of "Polyphenolic Hispolon Derived from Medicinal Mushrooms of the Inonotus and Phellinus Genera Promotes Wound Healing in Hyperglycemia-Induced Impairments"

_nutrients, 2025, doi:10.3390/nu17020266_

Round 1
Reviewer 1 Report
Comments and Suggestions for Authors
Authors report the wound healing effects of hispolon, which is derived from medicinal mushrooms. Promising results in the healing effects of polyphenolic hispolon were obtained in this work, although the detailed mechanism was not revealed.
1. Introduction: The hispolon may be not popular and familiar with the readers. Authors should add more information including the chemical structure, molecular weight, and chemical name. I only understood that the hispolon is polyphenolic compound.
2. The origin of the therapeutic effects of the hispolon was not shown. Authors should provide some hypothesis. I think that it may be came from the phenolic moieties due to its antioxidant activities. It requires a few references about the biological activities of phenolic compounds including polyphenols.
3. The chemical structure of hispolon must be provided in this manuscript.
Reviewer 2 Report
Comments and Suggestions for Authors
I recommend the authors:
- the conclusions chapter should be highlighted "5. Conclusions"
- the continuation of research in order to be able to introduce a new remedy for people affected by diabetes and to reduce the risk of severe illnesses or even death.
Reviewer 3 Report
Comments and Suggestions for Authors
Comments to the authors:
1- Please add the name of the mushroom to the title.
2- Adding graphical abstract is highly required.
3- In the abstract, authors need to provide a prisma under methodology in both in vitro and in vivo assays showing how assays were conducted, the used concentration of hispolon, and in which form it was used.
4- A quick scan of the literature showed various anti-diabetic assays on hispolon as recently cited in “Aanniz, T., Zeouk, I., Elouafy, Y., Touhtouh, J., Hassani, R., Hammani, K., ... & Bouyahya, A. (2024). Initial report on the multiple biological and pharmacological properties of hispolon: Exploring stochastic mechanisms. Biomedicine & Pharmacotherapy, 177, 117072.”, Authors should add the originality of their work?
5- What was the source of hispolon ointment or how it was formulated?
6- Authors could use more specific keywords.
7- In the introduction, please mention the amount of hispolon in mushroom, the recommended daily dose of mushroom, and the side effects/limitations of excessive administration.
8- Please mention the isolation and purification techniques of hispolon from mushroom.
9- Authors could benefit from this reference in the introduction part “Chen, Z., Wu, W., Wen, Y., Zhang, L., Wu, Y., Farid, M. S., ... & Zhao, C. (2023). Recent advances of natural pigments from algae. Food Production, Processing and Nutrition, 5(1), 39.”.
10- The authors would expand on the anti-diabetic activities of mushroom and hispolon and compare the findings with the previously published reports. Also, please highlight the suggested possible mechanisms of actions.
11- Please, mention the different formulations in which hispolon could be applied as an anti-diabetic agent.
12- Please, add hispolon chemical structure to the manuscript.
13- Lines 415-417: “Treatment with hispolon enhanced migration in high glucose incubated cells in a concentration-dependent manner”, please give more details about the concentrations used.
14- Lines 455-456: “Applying hispolon ointment at concentrations of 2.5% 455 (w/w) or 5% (w/w) to STZ-diabetic rats shortened the epithelialization time”, to which degree epithelialization time was shortened?
15- Please, make sure to mention the concentrations and the corresponding effects in each section of the results.
16- Could you put the conclusion in a separate section?
17- Please, add the future perspectives to the conclusion part.
Reviewer 4 Report
Comments and Suggestions for Authors
1. In the introduction of the article, please further specify the purpose of the research.
2. The authors should create a subsection of chemicals in the Materials and methods section. Please describe there all the main reagents used in the research including the companies where they were purchased. In particular, information on hipsolon - the main protagonist of the article was quite hidden in the methods. The reader cannot search through the entire article to find information on where the studied compound came from.
3. In vitro studies should be carried out on at least two cell lines.
4. What effect does hipsolon have on the survival of control cells after 12, 24 and 48 hours? Is it not toxic?
5. What were the levels of ROS, lipid peroxidation, 8-OHdG and GSH under NG conditions? Analogously as shown in the graph with survival rate. Did hipsolon alone at the concentrations tested affect the levels of these parameters tested in control cells without glucose addition?
6. Please clarify the concentrations of hipsolon used, values with different units appear (Figure 1B and 2C/D mmol/L, which seems to me to be a mistake) while in other places it is µmol/L.
7. The authors assessed the level of the low-molecular-weight antioxidant GSH. What is the oxidation of proteins, such as the level of carbonyl or amino groups?
8. The article is extensive, there are a lot of results. I understand that the authors wanted to put together in vitro and in vivo studies, but you might want to consider separating it into two parts. The paper needs quite deep analysis, not necessarily a lot but in general.
8. Why percentages of hipsolon are used in in vivo studies? Concentrations in mg/ kg body weight can be used.
9. For in vivo studies, analogous suggestion as in point 5 regarding cell lines. How does hipsolon alone in the same concentrations affect all tested parameters under control conditions?
10. Figures for in vivo studies are too blurry, they are unreadable because they contain a huge amount of data (reference to point 8). The figures need to be improved to make them clear and transparent to the audience.
11. Compared to the huge amount of data presented in the results section, the discussion is written too generally and needs to be corrected and supplemented.
Round 2
Reviewer 4 Report
Comments and Suggestions for Authors
After making the suggested changes to the manuscript and responding to my review comments, I believe that the article can be published.